# Efficient Riemannian Optimization on the Stiefel Manifold via the Cayley Transform

**Jun Li, Li Fuxin, Sinisa Todorovic**
School of EECS
Oregon State University
Corvallis, OR 97331
`{liju2,lif,sinisa}@oregonstate.edu`

## Abstract

Strictly enforcing orthonormality constraints on parameter matrices has been shown advantageous in deep learning. This amounts to Riemannian optimization on the Stiefel manifold, which, however, is computationally expensive. To address this challenge, we present two main contributions: (1) A new efficient retraction map based on an iterative Cayley transform for optimization updates, and (2) An implicit vector transport mechanism based on the combination of a projection of the momentum and the Cayley transform on the Stiefel manifold. We specify two new optimization algorithms: Cayley SGD with momentum, and Cayley ADAM on the Stiefel manifold. Convergence of Cayley SGD is theoretically analyzed. Our experiments for CNN training demonstrate that both algorithms: (a) Use less running time per iteration relative to existing approaches that enforce orthonormality of CNN parameters; and (b) Achieve faster convergence rates than the baseline SGD and ADAM algorithms without compromising the performance of the CNN. Cayley SGD and Cayley ADAM are also shown to reduce the training time for optimizing the unitary transition matrices in RNNs.

## 1 Introduction

Orthonormality has recently gained much interest, as there are significant advantages of enforcing orthonormality on parameter matrices of deep neural networks. For CNNs, Bansal et al. (2018) show that orthonormality constraints improve accuracy and gives a faster empirical convergence rate, Huang et al. (2018a) show that orthonormality stabilizes the distribution of neural activations in training, and Cogswell et al. (2015) show that orthonormality reduces overfitting and improves generalization. For RNNs, Arjovsky et al. (2016); Zhou et al. (2006) show that the orthogonal hidden-to-hidden matrix alleviates the vanishing and exploding-gradient problems.

Riemannian optimization on the Stiefel manifold, which represents the set of all orthonormal matrices of the same size, is an elegant framework for optimization under orthonormality constraints. But its high computational cost is limiting its applications, including in deep learning. Recent efficient approaches incorporate orthogonality in deep learning only for square parameter matrices (Arjovsky et al., 2016; Dorobantu et al., 2016), or indirectly through regularization (Bansal et al., 2018), which however does not guarantee the exact orthonormality of parameter matrices.

To address the aforementioned limitations, our first main contribution is an efficient estimation of the retraction mapping based on the Cayley transform for updating large *non-square* matrices of parameters on the Stiefel manifold. We specify an efficient iterative algorithm for estimating the Cayley transform that consists of only a few matrix multiplications, while the closed-form Cayley transform requires costly matrix inverse operations (Vorontsov et al., 2017; Wisdom et al., 2016). The efficiency of the retraction mapping is both theoretically proved and empirically verified in the paper.

Our second main contribution is aimed at improving convergence rates of training by taking into account the momentum in our optimization on the Stiefel manifold. We derive a new approach to move the tangent vector between tangent spaces of the manifold, instead of using the standard parallel transport. Specifically, we regard the Stiefel manifold as a submanifold of a Euclidean

space. This allows for representing the vector transport (Absil et al., 2009) as a projection onto the tangent space. As we show, since the Cayley transform implicitly projects gradients onto the tangent space, the momentum updates result in an implicit vector transport. Thus, we first compute a linear combination of the momentum and the gradient in the Euclidean space, and then update the network parameters using the Cayley transform, without explicitly performing the vector transport.

We apply the above two contributions to generalize the standard SGD with momentum and ADAM (Kingma & Ba, 2014) to the Stiefel manifold, resulting in our two new optimization algorithms called Cayley SGD with momentum and Cayley ADAM. A theoretical analysis of the convergence of Cayley SGD is presented. Similar analysis for Cayley ADAM is omitted, since it is very similar to the analysis presented in (Becigneul & Ganea, 2019).

Cayley SGD and Cayley ADAM are empirically evaluated on image classification using VGG and Wide ResNet on the CIFAR-10 and CIFAR-100 datasets (Krizhevsky & Hinton, 2009). The experiments show that Cayley SGD and Cayley ADAM achieve better classification performance and faster convergence rate than the baseline SGD with momentum and ADAM. While the baselines take less time per epoch since they do not enforce the orthonormality constraint, they take more epochs until convergence than Cayley SGD and Cayley ADAM. In comparison with existing optimization methods that also account for orthonormality – e.g., Polar decomposition, QR decomposition, or closed-form Cayley transform – our Cayley SGD and Cayley ADAM run much faster and yield equally good or better performance in image classification.

Finally, we apply the aforementioned two contributions to training of the unitary RNNs. Wisdom et al. (2016) proposes the full capacity unitary RNN that updates the hidden-to-hidden transition matrix with the closed-form Cayley transform. In contrast, for our RNN training, we use the more efficient Cayley SGD with momentum and Cayley ADAM. The results show that our RNN training takes less running time per iteration without compromising performance.

## 2 RELATED WORK

There is a host of literature on using orthonormality constraints in neural-network training. This section reviews closely related work, which can be broadly divided in two groups: orthonormality regularization and Riemannian optimization.

Regularization approaches can be divided as hard, which strictly enforce orthonormality, and soft. Hard regularizations are computationally expensive. For example, Huang et al. (2018b) extend Batch Normalization (Ioffe & Szegedy, 2015) with ZCA, and hence require costly eigenvalue decomposition. Huang et al. (2018a) derive a closed-form solution that also requires eigenvalue decomposition. Bansal et al. (2018); Cisse et al. (2017); Xiong et al. (2016) introduce mutual coherence regularization and spectral restricted isometry regularization; however, these regularizations are soft in that they can not guarantee orthonormality.

Riemannian optimization guarantees that the solution respects orthonormality constraints. For example, Cho & Lee (2017) replace Batch Normalization layers in a CNN with Riemannian optimization on the Grassmann manifold $G(n, 1)$, where the parameters are normalized but not orthonormalized. Also, Vorontsov et al. (2017); Wisdom et al. (2016); Lezcano-Casado & Martínez-Rubio (2019); Helfrich et al. (2017) perform Riemannian optimization on the group of unitary matrices to stablize RNNs training, but their technique cannot be applied to non-square parameter matrices. Becigneul & Ganea (2019) introduce a more general Riemannian optimization, but do not show how to efficiently perform this optimization on the Stiefel manifold.

The key challenge of Riemannian optimization is that exponential mapping — the standard step for estimating the next update point — is computationally expensive on the Stiefel manifold. Some methods use an efficient pseudo-geodesic retraction instead of the exponential mapping. For example, Absil & Malick (2012); Gawlik & Leok (2018); Manton (2002) use a projection-based method to map the gradient back to the Stiefel manifold that rely on computational expensive SVD. Other approaches are based on the closed-form Cayley transform (Fiori et al., 2012; Jiang & Dai, 2015; Nishimori & Akaho, 2005; Zhu, 2017), but require the costly matrix inversion. Also, Wen & Yin (2013) reduce the cost of the Cayley transform by making the restrictive assumption that the matrix size $n \times p$ is such that $2p \ll n$. Unfortunately, this algorithm is not efficient when $2p \geq n$.

## 3 PRELIMINARY

This section briefly reviews some well-known properties of the Riemannian and Stiefel manifolds. The interested reader is referred to Boothby (1986); Edelman et al. (1998) and references therein.

### 3.1 RIEMANNIAN MANIFOLD

**Definition 1.** *Riemannian Manifold: A Riemannian manifold $(\mathcal{M}, \rho)$ is a smooth manifold $\mathcal{M}$ equipped with a Riemannian metric $\rho$ defined as the inner product on the tangent space $T_x\mathcal{M}$ for each point $x$, $\rho_x(\cdot, \cdot) : T_x\mathcal{M} \times T_x\mathcal{M} \to \mathbb{R}$.*

**Definition 2.** *Geodesic, Exponential map and Retraction map: A geodesic is a locally shortest curve on a manifold. An exponential map, $Exp_x(\cdot)$, maps a tangent vector, $v \in T_x$, to a manifold, $\mathcal{M}$. $Exp_x(tv)$ represents a geodesic $\gamma(t) : t \in [0, 1]$ on a manifold, s.t. $\gamma(0) = x, \dot{\gamma}(0) = v$. A retraction map is defined as a smooth mapping $R_x : T_x\mathcal{M} \to \mathcal{M}$ on a manifold iff $R_x(0) = x$ and $DR_x(0) = id_{T_x\mathcal{M}}$, where $DR_x$ denotes the derivative of $R_x$, $id_{T_x\mathcal{M}}$ denotes an identity map defined on $T_x\mathcal{M}$. It is easy to show that any exponential map is a retraction map. As computing an exponential map is usually expensive, a retraction map is often used as an efficient alternative.*

**Definition 3.** *Parallel transport and vector transport: Parallel transport is a method to translate a vector along a geodesic on a manifold while preserving norm. A vector transport $\tau$ is a smooth map defined on a retraction $R$ of a manifold $\mathcal{M}$, $\tau : T_x\mathcal{M} \times T_x\mathcal{M} \to T_{R(\eta_x)}, (\eta_x, \xi_x) \mapsto \tau_{\eta_x}(\xi_x)$. $\tau$ satisfies the following properties: (1) Underlying retraction: $\tau_{\eta_x}(\xi_x) \in T_{R(\eta_x)}$; (2) Consistency: $\tau_{0_x}\xi_x = \xi_x, \forall \xi_x \in T_x\mathcal{M}$; (3) Linearity: $\tau_{\eta_x}(a\xi_x + b\zeta_x) = a\tau_{\eta_x}(\xi_x) + b\tau_{\eta_x}(\zeta_x)$. Usually, a vector transport is a computationally efficient alternative to a parallel transport.*

### 3.2 STIEFEL MANIFOLD

The Stiefel manifold $\mathrm{St}(n, p)$, $n \geq p$, is a Riemannian manifold that is composed of all $n \times p$ orthonormal matrices $\{X \in \mathbb{R}^{n \times p} : X^T X = I\}$. In the rest of the paper, we will use notation $\mathcal{M} = \mathrm{St}(n, p)$ to denote the Stiefel manifold. We regard $\mathcal{M}$ as an embeded submanifold of a Euclidean space. Hence, the Riemannian metric $\rho$ is the Euclidean metric as: $\rho_X(Z_1, Z_2) = tr(Z_1^\top Z_2)$, where $Z_1, Z_2$ are tangent vectors in $T_X\mathcal{M}$. The tangent space of $\mathcal{M}$ at $X$ is defined as

$$T_X\mathcal{M} = \{Z : Z^\top X + X^\top Z = 0\} \tag{1}$$

The projection of a matrix $Z \in \mathbb{R}^{n \times p}$ to $T_X\mathcal{M}$ can be computed as

$$\pi_{T_X}(Z) = WX$$
$$\text{where: } W = \hat{W} - \hat{W}^\top, \quad \hat{W} = ZX^\top - \frac{1}{2}X(X^\top ZX^\top). \tag{2}$$

Given a derivative of the objective function $\nabla f(X)$ at $X$ in the Euclidean space, we can compute the Riemannian gradient $\nabla_\mathcal{M}f(X)$ on the Stiefel manifold as a projection onto $T_X\mathcal{M}$ using $\pi_{T_X}(\nabla f(X))$ given by Eq.(2). It follows that optimization of $f$ on the Riemannian manifold can be computed as follows. First, compute $\nabla_\mathcal{M}f(X_t)$ in $T_X\mathcal{M}$. Second, transport the momentum $M_t$ to the current tangent space $T_X\mathcal{M}$ and combine it linearly with the current Riemannian gradient $\nabla_\mathcal{M}f(X_t)$ to update the momentum $M_{t+1}$. Finally, third, update the new parameter $X_{t+1}$ along a curve on the manifold with the initial direction as $M_{t+1}$.

While the exponential map and parallel transport can be used to update parameters and momentums in optimization on the Riemannian manifold, they are computationally infeasible on the Stiefel manifold. In the following section, we specify our computationally efficient alternatives.

#### 3.2.1 PARAMETER UPDATES BY ITERATIVE CAYLEY TRANSFORM

The Cayley transform computes a parametric curve on the Stiefel manifold using a skew-symmetric matrix (Nishimori & Akaho, 2005). The closed-form of the Cayley transform is given by:

$$Y(\alpha) = (I - \frac{\alpha}{2}W)^{-1}(I + \frac{\alpha}{2}W)X, \tag{3}$$

where $W$ is a skew-symmetric matrix, i.e. $W^\top = -W$, $X$ is on the Stiefel manifold, and $\alpha \geq 0$ is a parameter that represents the length on the curve. It is straightforward to verify that

$$Y(0) = X \quad and \quad Y^{'}(0) = WX. \tag{4}$$

From Definition 2 and the definition of the tangent space of the Stiefel manifold given by Eq.(1), the Cayley transform is a valid retraction map on the Stiefel manifold. By choosing $W = \hat{W} - \hat{W}^\top$, where $\hat{W} = \nabla f(X)X^\top - \frac{1}{2}X(X^\top \nabla f(X)X^\top)$ as in Eq.(2), we see that the Cayley transform implicitly projects gradient on the tangent space as its initial direction. Therefore, the Cayley transform can represent an update for the parameter matrices on the Stiefel manifold.

However, the closed-form Cayley transform in Eq.(3) involves computing the expensive matrix inversion, which cannot be efficiently performed in large deep neural networks.

Our first contribution is an iterative estimation of the Cayley transform that efficiently uses only matrix multiplications, and thus is more efficient than the closed form in Eq.(3). We represent the Cayley transform with the following fixed-point iteration:

$$Y(\alpha) = X + \frac{\alpha}{2}W\left(X + Y(\alpha)\right), \tag{5}$$

which can be proved by moving $Y(\alpha)$ on the right-hand side to the left-hand side in Eq.(3). The expression in Eq.(5) is an efficient approximation of Eq.(3). In Sec. 5, we will analyze its convergence rate to the closed-form Eq.(3).

### 3.2.2 MOMENTUM UPDATES BY THE IMPLICIT VECTOR TRANSPORT

Our second contribution is an efficient way to perform momentum updates on the Stiefel manifold. We specify an implicit vector transport by combining the Cayley transform and momentum updates in an elegant way without explicitly computing the vector transport. As the Stiefel manifold can be viewed as a submanifold of the Euclidean space $\mathbb{R}$, we have a natural inclusion of the tangent space $T_X\mathcal{M} \subset \mathbb{R}$. Consequently, the vector transport on the Stiefel manifold is the projection on the tangent space. Formally, for two tangent vectors $\xi_X, \eta_X \in T_X\mathcal{M}$, the vector transport of $\xi_X$ along a retraction map $r(\eta_X)$, denoted as $\tau_{\eta_X}(\xi_X)$, can be computed as:

$$\tau_{\eta_X}(\xi_X) = \pi_{T_{r(\eta_X)}}(\xi_X). \tag{6}$$

We specify the retraction map $r(\cdot)$ as the Cayley transform $Y$ in Eq.(3). At optimization step $k$, in Eq.(6), we choose $\xi_X = \eta_X = M_k$, where $M_k$ is the momentum in step $k-1$. Then, we compute the vector transport of $M_k$ as $\tau_{M_k}(M_k) = \pi_{T_{X_k}}(M_k)$. As the projection onto a tangent space is a linear map, then $\alpha\tau_{M_k}(M_k) + \beta\nabla_\mathcal{M}f(X_k) = \alpha\pi_{T_{X_k}}(M_k) + \beta\pi_{T_{X_k}}(\nabla f(X_k)) = \pi_{T_{X_k}}(\alpha M_k + \beta\nabla f(X_k))$. Thus we first compute a linear combination of the Euclidean gradient $\nabla f(X)$ and the momentum $M_k$, as in the Euclidean space, and then use the iterative Cayley transform to update the parameters, without explicitly estimating the vector transport, since the Cayley transform implicitly project a vector onto the tangent space.

## 4 ALGORITHMS

This section specifies our Cayley SGD and Cayley ADAM algorithms. Both represent our efficient Riemannian optimization on the Stiefel manifold that consists of two main steps. As the Cayley transform implicitly projects gradient and momentum vectors onto the tangent space, we first linearly combine the momentum of the previous iteration with the stochastic gradient of the objective function $f$ at the current point $X$, denoted as $\mathcal{G}(X)$. Then, we use the iterative Cayley transform to estimate the next optimization point based on the updated momentum. This is used to generalize the conventional SGD with momentum and ADAM algorithms to our two new Riemannian optimizations on the Stiefel manifold, as described in Section 4.1 and Section 4.2.

---

**Algorithm 1** Cayley SGD with Momentum

---

1: **Input:** learning rate $l$, momentum coefficient $\beta$, $\epsilon=10^{-8}$, $q = 0.5$, $s = 2$.
2: Initialize $X_1$ as an orthonormal matrix; and $M_1 = 0$
3: **for** $k = 0$ **to** $T$ **do**
4:     $M_{k+1} \leftarrow \beta M_k - \mathcal{G}(X_k)$,                             $\triangleright$ Update the momentum
5:     $\hat{W}_k \leftarrow M_{k+1}X_k^\top - \frac{1}{2}X_k(X_k^\top M_{k+1}X_k^\top)$         $\triangleright$ *Compute the auxiliary matrix*
6:     $W_k \leftarrow \hat{W}_k - \hat{W}_k^\top$
7:     $M_{k+1} \leftarrow W_k X_k$.               $\triangleright$ *Project momentum onto the tangent space*
8:     $\alpha \leftarrow \min\{l, 2q/(\|W_k\| + \epsilon)\}$     $\triangleright$ *Select adaptive learning rate for contraction mapping*
9:     Initialize $Y^0 \leftarrow X + \alpha M_{k+1}$              $\triangleright$ *Iterative estimation of the Cayley Transform*
10:     **for** $i = 1$ **to** $s$ **do**
11:         $Y^i \leftarrow X_k + \frac{\alpha}{2}W_k(X_k + Y^{i-1})$
12:     Update $X_{k+1} \leftarrow Y^s$

---

**Algorithm 2** Cayley ADAM

---

1: **Input:** learning rate $l$, momentum coefficients $\beta_1$ and $\beta_2$, $\epsilon = 10^{-8}$, $q = 0.5$, $s = 2$.
2: Initialize $X_1$ as an orthonormal matrix. $M_1 = 0$, $v_1 = 1$
3: **for** $k = 0$ **to** $T$ **do**
4:     $M_{k+1} \leftarrow \beta_1 M_k + (1 - \beta_1)\mathcal{G}(X_k)$             $\triangleright$ *Estimate biased momentum*
5:     $v_{k+1} \leftarrow \beta_2 v_k + (1 - \beta_2)\|\mathcal{G}(X_k)\|^2$
6:     $\hat{v}_{k+1} \leftarrow v_{k+1}/(1 - \beta_2^k)$            $\triangleright$ *Update biased second raw moment estimate*
7:     $r \leftarrow (1 - \beta_1^k)\sqrt{\hat{v_{k+1}} + \epsilon}$               $\triangleright$ *Estimate biased-corrected ratio*
8:     $\hat{W}_k \leftarrow M_{k+1}X_k^\top - \frac{1}{2}X_k(X_k^\top M_{k+1}X_k^\top)$   $\triangleright$ *Compute the auxiliary skew-symmetric matrix*
9:     $W_k \leftarrow (\hat{W}_k - \hat{W}_k^\top)/r$
10:     $M_{k+1} \leftarrow r W_k X_k$               $\triangleright$ *Project momentum onto the tangent space*
11:     $\alpha \leftarrow \min\{l, 2q/(\|W_k\| + \epsilon)\}$   $\triangleright$ *Select adaptive learning rate for contraction mapping*
12:     Initialize $Y^0 \leftarrow X_k - \alpha M_{k+1}$         $\triangleright$ *Iterative estimation of the Cayley Transform*
13:     **for** $i = 1$ **to** $s$ **do**
14:         $Y^i \leftarrow X_k - \frac{\alpha}{2}W(X_k + Y^{i-1})$
15:     Update $X_{k+1} \leftarrow Y^s$

---

### 4.1 CAYLEY SGD WITH MOMENTUM

We generalize the heavy ball (HB) momentum update (Ghadimi et al., 2015; Zavriev & Kostyuk, 1993) in the $k$th optimization step[1] to the Stiefel manifold. Theoretically, it can be represented as:

$$M_{k+1} = \beta\pi_{\mathcal{T}_{X_k}}(M_k) - \mathcal{G}_\mathcal{M}(X_k), \quad X_{k+1} = Y(\alpha, X_k, W_k) \tag{7}$$

where $Y(\alpha, X_k, W_k)$ is a curve that starts at $X_k$ with length $\alpha$ on the Stiefel manifold, specified by the Cayley transform in Eq.(3). In practice, we efficiently perform the updates in Eq.(7) by the proposed iterative Cayley transform and implicit vector transport on the Stiefel manifold. Specially, we first update the momentum as if it were in the Euclidean space. Then, we update the new parameters by iterative Cayley transform. Finally, we correct the momentum $M_{k+1}$ by projecting it to $T_{X_k}\mathcal{M}$. The details of our Cayley SGD algorithm are shown in Alg. 1.

### 4.2 ADAM ON THE STIEFEL MANIFOLD

ADAM is a recent first-order optimization method for stochastic objective functions. It estimates adaptive lower-order moments and uses adaptive learning rate. The algorithm is designed to combine the advantages of both AdaGrad, which performs well in sparse-gradient cases, and RMSProp, which performs well in non-stationary cases.

We generalize ADAM to the Stiefel manifold by making three modifications to the vanilla ADAM. First, we replace the standard gradient and momentum with the corresponding ones on the Stiefel manifold, as described in Section 4.1. Second, we use a manifold-wise adaptive learning rate that

---

[1]Note that we use index $i$ in superscript to indicate our iterative steps in estimation of the Cayley transform, and index $k$ in subscript to indicate optimization steps.

assign a same learning rate for all entries in a parameter matrix as in (Absil et al., 2009). Third, we use the Cayley transform to update the parameters. Cayley ADAM is summarized in Alg 2.

## 5 CONVERGENCE ANALYSIS

In this section, we analyze convergence of the iterative Cayley transform and Cayley SGD with momentum. To facilitate our analysis, we make the following common assumption.

**Assumption 1.** *The gradient $\nabla f$ of the objective function $f$ is Lipschitz continuous*

$$\|\nabla f(X) - \nabla f(Y)\| \leq L\|X - Y\|, \quad \forall X, Y, \text{ where } L > 0 \text{ is a constant.} \tag{8}$$

Lipschitz continity is widely applicable to deep learning architectures. For some models using ReLU, the derivative of ReLU is Lipschitz continuous almost everywhere with an appropriate Lipschitz constant $L$ in Assumption 1 , except for a small neighbourhood around 0, whose measure tends to 0. Such cases do not affect either analysis in theory or training in practice.

The above assumption allows proving the following contraction mapping theorem.

**Theorem 1.** *For $\alpha \in (0, \min\{1, \frac{2}{\|W\|}\})$, the iteration $Y^{i+1} = X + \frac{\alpha}{2}W\left(X + Y^i\right)$ is a contraction mapping and converges to the closed-form Cayley transform $Y(\alpha)$ given by Eq.(3). Specifically, at iteration $i$, $\|Y^i - Y(\alpha)\| = o(\alpha^{2+i})$.*

Theorem 1 shows the iterative Cayley transform will converge. Specially, it converges faster than other approximation algorithms, such as, e.g., the Newton iterative and Neumann Series which achieves error bound $o(\alpha^i)$ at the $i^{th}$ iteration. We further prove the following result on convergence:

**Theorem 2.** *Given an objective function $f(X)$ that satisfies Assumption 1, let Cayley SGD with momentum run for $t$ iterations with $\mathcal{G}(X_k)$. For $\alpha = \min\{\frac{1-\beta}{L}, \frac{A}{\sqrt{t+1}}\}$, where $A$ is a positive constant, we have: $\min_{k=0,\cdots,t} E[\|\nabla_{\mathcal{M}} f(X_k)\|^2] = o(\frac{1}{\sqrt{t+1}}) \rightarrow 0, \text{ as } t \rightarrow \infty$.*

The proofs of Theorem 1 and Theorem 2 are presented in the appendix.

## 6 EXPERIMENTS

### 6.1 ORTHONORMALITY IN CNNS

In CNNs, for a convolutional layer with kernel $\hat{K} \in \mathbb{R}^{c_{out} \times c_{in} \times h \times w}$, we first reshape $\hat{K}$ into a matrix $K$ of size $p \times n$, where $p = c_{out}, n = c_{in} \times h \times w$. In most cases, we have $p \leq n$. Then, we restrict the matrix $K$ on the Stiefel manifold using Cayley SGD or Cayley ADAM, while other parameters are optimized with SGD and ADAM.

**Datasets:** We evaluate Cayley SGD or Cayley ADAM in image classification on the CIFAR10 and CIFAR100 datasets (Krizhevsky & Hinton, 2009). CIFAR10 and CIFAR100 consist of of 50,000 training images and 10,000 test images, and have 10 and 100 mutually exclusive classes.

**Models:** We use two networks — VGG (Simonyan & Zisserman, 2014) and Wide ResNet (Zagoruyko & Komodakis, 2016) — that obtain state of the art performance on CIFAR10 and CIFAR100. For VGG, every convolutional layer is followed by a batch normalization layer and a ReLU. For Wide ResNet, we use basic blocks, where two consecutive 3-by-3 convolutional layers are followed by the batch normalization and ReLU activation, respectively.

**Training Strategies:** We use different learning rates $l_e$ and $l_{st}$ for weights on the Euclidean space and the Stiefel manifold, respectively. We set the weight decay as 0.0005, momentum as 0.9, and minibatch size as 128. The initial learning rates are set as $l_e = 0.01$ and $l_{st} = 0.2$ for Cayley SGD and $l_e = 0.01$ and $l_{st} = 0.4$ for Cayley ADAM. During training, we reduce the learning rates by a factor of 0.2 at 60, 120, and 160 epochs. The total number of epochs in training is 200. In training, the data samples are normalized using the mean and variance of the training set, and augmented by randomly flipping training images.

**Our baselines** include SGD with momentum and ADAM. We follow the same training strategies as mentioned above, except for the initial learning rates set to 0.1 and 0.001, respectively.

Table 1: Classification errors(%) on CIFAR10.

| METHOD | VGG-13 | VGG-16 | VGG-19 | WRN 52-1 | WRN 16-4 | WRN 28-10 |
|---|---|---|---|---|---|---|
| SGD | 5.88 | 6.32 | 6.49 | 6.23 | 4.96 | 3.89 |
| ADAM | 6.43 | 6.61 | 6.92 | 6.77 | 5.32 | 3.86 |
| CAYLEY SGD | 5.90 | **5.77** | 5.85 | 6.35 | 5.15 | 3.66 |
| CAYLEY ADAM | 5.93 | 5.88 | 6.03 | 6.44 | 5.22 | **3.57** |

Table 2: Classification errors(%) on CIFAR100.

| METHOD | VGG-13 | VGG-16 | VGG-19 | WRN 52-1 | WRN 16-4 | WRN 28-10 |
|---|---|---|---|---|---|---|
| SGD | 26.17 | 26.84 | 27.62 | 27.44 | 23.41 | 18.66 |
| ADAM | 26.58 | 27.10 | 27.88 | 27.89 | 24.45 | 18.45 |
| CAYLEY SGD | **24.86** | 25.48 | 25.68 | 27.64 | 23.71 | 18.26 |
| CAYLEY ADAM | 25.10 | 25.61 | 25.70 | 27.91 | 24.18 | **18.10** |

**Performance:** Table 1 and Table 2 show classification errors on CIFAR10 and CIFAR100 respectively using different optimization algorithms. As shown in the tables, the proposed two algorithms achieve competitive performance, and for certain deep architectures, the best performance. Specifically, the network WRN-28-10 trained with Cayley ADAM achieves the best error rate of $3.57\%$ and $18.10\%$ on CIFAR10 and CIFAR100 respectively. Fig. 1 compares training curves of our algorithms and baselines in terms of epochs, and shows that both Cayley SGD and Cayley ADAM converge faster than the baselines. In particular, the training curves of the baselines tend to get stuck in a plateau before the learning rate drops, which is not the case with our algorithms. This might be because the baselines do not enforce orthonormality of network parameters. In training, the backpropagation of orthonormal weight vectors, in general, does not affect each other, and thus has greater chances to explore new parameter regions. Fig. 2 also compares the training loss curve in terms of time. Our Cayley SGD and Cayley ADAM converge the fastest among methods that also address orthonormality. Although the baselines SGD and ADAM converge faster at the beginning due to their training efficiency, our Cayley SGD and Cayley ADAM can catch up with the baseline after 12000 seconds, which corresponds to the 120th epoch of SGD and ADAM, and the 60th epoch of Cayley SGD and Cayley ADAM.

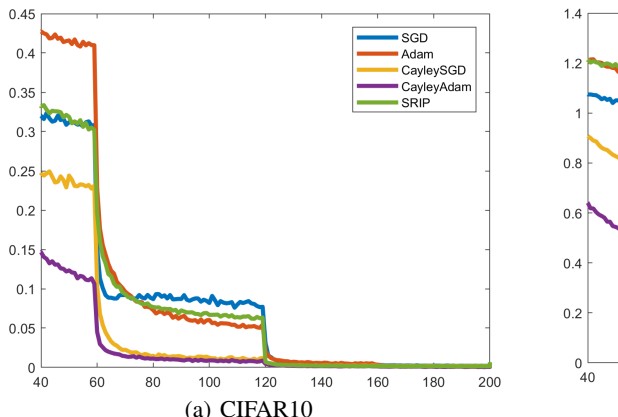
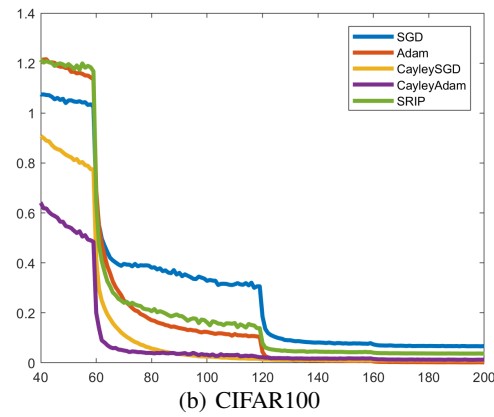

(a) CIFAR10      (b) CIFAR100

Figure 1: Training loss curves of different optimization algorithms for WRN-28-10 for epoach 40-200. (a) Results on CIFAR10. (b) Results on CIFAR100. Both figures show that our Cayley SGD and Cayley ADAM achieve the top two fastest convergence rates.

**Comparison with State of the Art.** We compare the proposed algorithms with two sets of state of the art. The first set of approaches are soft orthonormality regularization approaches (Bansal et al., 2018). Specially, for a weight matrix $K \in \mathbb{R}^{n \times p}$, SO penalizes $||KK^\top - I||_F^2$, DSO penalizes $(||KK^\top - I||_F^2 + ||K^\top K - I||_F^2)$, the SRIP penalizes the spectral norm of $(KK^\top - I)$. The second set of approaches includes the following hard orthonormality methods: Polar decomposition(Absil et al., 2009), QR decomposition(Absil et al., 2009), closed-form Cayley transform, Wen&Yin (Wen & Yin, 2013), OMDSM(Huang et al., 2018a), DBN(Huang et al., 2018b). Note that we do not in-

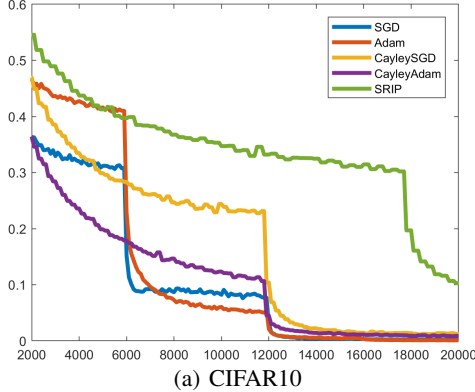
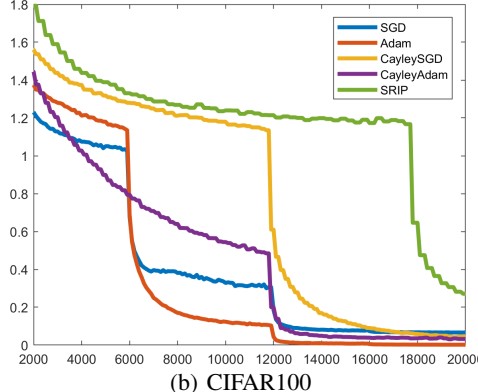

|(a) CIFAR10|(b) CIFAR100|

Figure 2: Training loss curves of different optimization algorithms for WRN-28-10 in terms of seconds. (a) Results on CIFAR10. (b) Results on CIFAR100.

clude momentum in Polar decomposition and QR decomposition as previous work does not specify the momentum. Also, we use the closed-form Cayley transform without momentum as an ablation study of the momentum effect. All experiments are evaluated on the benchmark network WRN-28-10. Table 3 shows that our algorithms achieve comparable error rates with state of the art. All algorithms are run on one TITAN Xp GPU, and their average training time are compared per epoch. Table 3 shows that our algorithms run much faster than existing algorithms, except for the baseline SGD and ADAM which do not impose orthonormality constraints.

## 6.2 ORTHONORMALITY IN RNNs

In RNNs, the hidden-to-hidden transition matrix $K$ can be modeled as a unitary matrix (Arjovsky et al., 2016). Wisdom et al. (2016) model the hidden-to-hidden transition matrix as a full-capacity unitary matrix on the complex Stiefel manifold: $\text{St}(\mathbb{C}^N) = \{K \in \mathbb{C}^{N \times N} : K^H K = I\}$.

**Pixel-by-pixel MNIST:** We evaluate the proposed algorithms on the challenging pixel-by-pixel MNIST task for long-term memory. The task was used to test the capacity of uRNNs (Arjovsky et al., 2016; Wisdom et al., 2016). Following the same setting as in Wisdom et al. (2016), we reshape MNIST images of $28 \times 28$ pixels to a $T = 784$ pixel-by-pixel sequences, and select 5,000 out of the 60,000 training examples for the early stopping validation.

**Training:** Wisdom et al. (2016) restricted the transition unitary matrix on the Stiefel manifold via a closed-form Cayley transform. On the contrary, we use Cayley SGD with momentum and Cayley ADAM to reduce the training time. Table 4 shows that the proposed algorithms reduce the training

Table 3: Error rate and training time per epoch comparison to baselines with WRN-28-10 on CIFAR10 and CIFAR100. All experiments are performed on one TITAN Xp GPU.

| | Method | Error Rate(%) | | Training time(s) |
|---|---|---|---|---|
| | | CIFAR10 | CIFAR100 | |
| Baselines | SGD | 3.89 | 18.66 | 102.5 |
| | ADAM | 3.85 | 18.52 | 115.2 |
| Soft orthonormality | SO (Bansal et al., 2018) | 3.76 | 18.56 | 297.3 |
| | DSO (Bansal et al., 2018) | 3.86 | 18.21 | 311.0 |
| | SRIP (Bansal et al., 2018) | 3.60 | 18.19 | 321.8 |
| Hard orthonormality | OMDSM (Huang et al., 2018a) | 3.73 | 18.61 | 943.6 |
| | DBN (Huang et al., 2018b) | 3.79 | 18.36 | 889.4 |
| | Polar (Absil et al., 2009) | 3.75 | 18.50 | 976.5 |
| | QR (Absil et al., 2009) | 3.75 | 18.65 | 469.3 |
| | Wen&Yin (Wen & Yin, 2013) | 3.82 | 18.70 | 305.8 |
| | Cayley closed form w/o momentum | 3.80 | 18.68 | 1071.5 |
| | Cayley SGD (**Ours**) | 3.66 | 18.26 | 218.7 |
| | Cayley ADAM (**Ours**) | 3.57 | 18.10 | 224.4 |

Table 4: Pixel-by-pixel MNIST accuracy and training time per iteration of the closed-form Cayley Transform, Cayley SGD, and Cayley ADAM for Full-uRNNs (Wisdom et al., 2016). All experiments are performed on one TITAN Xp GPU.

| Model | Hidden Size | Closed-Form | | Cayley SGD | | Cayley ADAM | |
|---|---|---|---|---|---|---|---|
| | | Acc(%) | Time(s) | Acc(%) | Time(s) | Acc(%) | Time(s) |
| Full-uRNN | 116 | 92.8 | 2.10 | 92.6 | 1.42 | 92.7 | 1.50 |
| Full-uRNN | 512 | 96.9 | 2.44 | 96.7 | 1.67 | **96.9** | 1.74 |

time by about 35% for all settings of the network, while maintaining the same level of accuracy. All experiments are performed on one TITAN Xp GPU.

**Checking Unitariness:** To show that the proposed algorithms are valid optimization algorithms on the Stiefel manifold, we check the unitariness of the hidden-to-hidden matrix $K$ by computing the error term $||K^H K - I||_F$ during training. Table 5 compares average errors for varying numbers of iterations $s$. As can be seen, the iterative Cayley transform can approximate the unitary matrix when $s = 2$. The iterative Cayley transform performs even better than the closed form Cayley transform, which might be an effect of the rounding error of the matrix inversion as in Eq.(3).

Table 5: Checking unitariness by computing the error $||K^H K - I||_F$ for varying numbers of iterations in the iterative Cayley transform and the closed-form Cayley transform.

| Hidden Size | s=0 | s=1 | s=2 | s=3 | s=4 | Closed-form |
|---|---|---|---|---|---|---|
| n=116 | 3.231e-3 | 2.852e-4 | 7.384e-6 | 7.353e-6 | 7.338e-6 | 8.273e-5 |
| n=512 | 6.787e-3 | 5.557e-4 | 2.562e-5 | 2.547e-5 | 2.544e-5 | 3.845e-5 |

## 7 CONCLUSION

We proposed an efficient way to enforce the exact orthonormal constraints on parameters by optimization on the Stiefel manifold. The iterative Cayley transform was applied to the conventional SGD and ADAM for specifying two new algorithms: Cayley SGD with momentum and Cayley ADAM, and the theoretical analysis of convergence of the former. Experiments show that both algorithms achieve comparable performance and faster convergence over the baseline SGD and ADAM in training of the standard VGG and ResNet on CIFAR10 and CIFAR100, as well as RNNs on the pixel-by-pixel MNIST task. Both Cayley SGD with momentum and Cayley ADAM take less runtime per iteration than all existing hard orthonormal methods and soft orthonormal methods, and can be applied to non-square parameter matrices.

ACKNOWLEDGEMENTS

This work was supported in part by NSF grant IIS-1911232, DARPA XAI Award N66001-17-2-4029 and AFRL STTR AF18B-T002.

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

## A  PRELIMINARY

In this section, we derive some properties of the Stiefel manifold in this section to facilitate the proofs of **Theorem 1** and **Theorem 2** in the main paper.

Considering the Stiefel manifold $\mathcal{M}$ is a bounded set and Lipschitz Assumption 1 on the gradient in Sec. 5, one straightforward conclusion is that both $\nabla f(X)$ and its Stiefel manifold gradient $\nabla_{\mathcal{M}} f(X)$ that is a projection onto the tangent space are bounded. Formally, there exists a positive constant $G$, such that

$$||\nabla_{\mathcal{M}} f(X)|| \leq ||\nabla f(X)|| \leq G, \forall X \in \mathcal{M} \tag{9}$$

As stochastic gradient $\mathcal{G}(X) = \mathcal{G}(X; \xi)$ is the gradient of a sub-dataset, where $\xi$ is a stochastic variable for data samples and we are working are a finite dataset, it's straightforward to show that $\mathcal{G}(X)$ and its Riemannian stochastic gradient $\mathcal{G}_{\mathcal{M}}(X)$ are also bounded. For brevity, we still use the same upper bound $G$, such that:

$$||\mathcal{G}_{\mathcal{M}}(X)|| \leq ||\mathcal{G}(X)|| \leq G, \forall X \in \mathcal{M} \tag{10}$$

Recall the recursion in Eq.(7), we show that the momentum is also bounded:

$$||M_{k+1}|| \leq \beta ||M_k|| + ||\mathcal{G}_{\mathcal{M}}(X_k)||$$

$$\leq \sum_{i=0}^{k} \beta^{k-i} ||\mathcal{G}_{\mathcal{M}}(X_i)||$$

$$\leq \frac{1}{1-\beta} G \tag{11}$$

Therefore, we know that $W_k$ in Eq.(7) is bounded.

## B  PROOF OF THEOREM 1

*Proof.* By subtracting the iterative relationship Eq.(5) by its $i^{th}$ iteration $Y^{i+1} = X + \frac{\alpha}{2} W (X + Y^i)$, we have:

$$||Y^{i+1} - Y(\alpha)|| \leq \frac{\alpha ||W||}{2} ||Y^i - Y(\alpha)|| \tag{12}$$

Therefore, since $W$ is bounded, for $\alpha < \frac{2}{||W||}$, such that $\frac{\alpha ||W||}{2} < 1$, the iteration in Eq.(5) is a contraction mapping, and it will converge to the closed-from solution $Y(\alpha)$.

By differentiate Eq.(5), we have:

$$\frac{dY(\alpha)}{d\alpha} = W(\frac{X + Y(\alpha)}{2}) + \frac{\alpha}{2} W \frac{dY(\alpha)}{d\alpha}$$

$$\frac{d^2 Y(\alpha)}{d\alpha^2} = (I - \frac{\alpha}{2} W)^{-1} W \frac{dY(\alpha)}{d\alpha} \tag{13}$$

therefore, $\frac{dY(\alpha)}{d\alpha}$ and $\frac{d^2 Y(\alpha)}{d\alpha^2}$ are bounded, i.e. there exist a positive constant $C$, such that:

$$||\frac{d^2 Y(\alpha)}{d\alpha^2}|| \leq C \tag{14}$$

Using the Taylor expansion of $Y$ in Eq.(3), we have:

$$Y(\alpha) = X_k + \alpha M_{k+1} + \frac{1}{2} \alpha^2 \frac{d^2 Y(\gamma_k \alpha)}{d\alpha^2} \tag{15}$$

where $\gamma_k \in (0, 1)$. Given $Y^0 = X_k + \alpha M_{k+1}$, we have:

$$||Y^0 - Y(\alpha)|| = o(\alpha^2) \tag{16}$$

Since $W$ is bounded and $\frac{\alpha ||W||}{2} < 1$, then,

$$||Y^i - Y(\alpha)|| \leq (\frac{\alpha ||W||}{2})^i ||Y^0 - Y(\alpha)|| = o(\alpha^{2+i}) \tag{17}$$

$\square$

## C PROOF OF THEOREM 2

*Proof.* Use Taylor expansion of $Y(\alpha)$, the process of Cayley SGD with momentum Eq.(7) can be written as:

$$M_{k+1} = \pi_{\mathcal{T}_{X_k}}(\beta M_k) - \mathcal{G}_{\mathcal{M}}(X_k)$$

$$X_{k+1} = X_k + \alpha M_{k+1} + \frac{1}{2}\alpha^2 \frac{d^2 Y(\gamma_k \alpha)}{d\alpha^2} \tag{18}$$

where $\gamma_k \in (0, 1)$.

Using the fact that

$$M = \pi_{\mathcal{T}_X}(M) + \pi_{\mathcal{N}_X}(M) \tag{19}$$

where $\pi_{\mathcal{N}_X}(M)$ is the projection onto the normal space, and

$$\pi_{\mathcal{N}_X}(M) = X\frac{X^\top M + M^\top X}{2} \tag{20}$$

Then, the projection of momentum can be represented as:

$$\pi_{\mathcal{T}_{X_k}}(M_k) = M_k - \pi_{\mathcal{N}_{X_k}}(M_k)$$

$$= M_k - X_k\frac{X_k^\top M_k + M_k^\top X_k}{2}$$

$$= M_k - \frac{1}{2}X_k\{[X_{k-1} + \alpha M_k + \frac{1}{2}\alpha^2 \frac{d^2 Y(\gamma_{k-1}\alpha)}{d\alpha^2}]^\top M_k$$

$$+ M_k^\top[X_{k-1} + \alpha M_k + \frac{1}{2}\alpha^2 \frac{d^2 Y(\gamma_{k-1}\alpha)}{d\alpha^2}]\}$$

$$= M_k - \alpha X_k M_k^\top M_k - \frac{1}{4}\alpha^2 X_k[\frac{d^2 Y(\gamma_{k-1}\alpha)}{d\alpha^2}^\top M_k + M_k^\top \frac{d^2 Y(\gamma_{k-1}\alpha)}{d\alpha^2}] \tag{21}$$

Then the momentum update in Eq.(7) is equivalent to:

$$M_{k+1} = \beta M_k - \alpha\beta X_k M_k^\top M_k - \mathcal{G}_{\mathcal{M}}(X_k)$$

$$- \frac{1}{4}\alpha^2\beta X_k[\frac{d^2 Y(\gamma_{k-1}\alpha)}{d\alpha^2}^\top M_k + M_k^\top \frac{d^2 Y(\gamma_{k-1}\alpha)}{d\alpha^2}] \tag{22}$$

Therefore, the paramter update in Eq.(7) can be represented as:

$$X_{k+1} = X_k + \alpha\beta M_k - \alpha\mathcal{G}_{\mathcal{M}}(X_k)$$

$$- \frac{1}{4}\alpha^3\beta X_k[\frac{d^2 Y(\gamma_{k-1}\alpha)}{d\alpha^2}^\top M_k + M_k^\top \frac{d^2 Y(\gamma_{k-1}\alpha)}{d\alpha^2}]$$

$$- \alpha^2\beta X_k M_k^\top M_k + \frac{1}{2}\alpha^2 \frac{d^2 Y(\gamma_k\alpha)}{d\alpha^2}$$

$$= X_k + \beta(X_k - X_{k-1}) - \alpha\mathcal{G}_{\mathcal{M}}(X_k) + \alpha^2 U \tag{23}$$

where

$$U = -\frac{1}{4}\alpha\beta X_k[\frac{d^2 Y(\gamma_{k-1}\alpha)}{d\alpha^2}^\top M_k + M_k^\top \frac{d^2 Y(\gamma_{k-1}\alpha)}{d\alpha^2}]$$

$$- \beta X_k M_k^\top M_k + \frac{1}{2}\frac{d^2 Y(\gamma_k\alpha)}{d\alpha^2} - \frac{\beta}{2}\frac{d^2 Y(\gamma_{k-1}\alpha)}{d\alpha^2} \tag{24}$$

Since $||M||$, $||X||$, and $||\frac{d^2 Y}{d\alpha^2}||$ are bounded, there is a positive constant $D$, such that

$$||U|| \le D \tag{25}$$

To facilitate the analysis of Cayle SGD with momentum, we introduce auxiliary variables $\{P_k\}$, such that:

$$Z_{k+1} = Z_k - \frac{\alpha}{1-\beta}\mathcal{G}_{\mathcal{M}}(X_k) + \frac{\alpha^2}{1-\beta}U \tag{26}$$

where

$$Z_k = X_k + P_k \tag{27}$$

and

$$P_k = \begin{cases} \frac{\beta}{1-\beta}(X_k - X_{k-1}), k \geq 1 \\ 0, k = 0 \end{cases}$$

Since $f(X)$ is a smooth function according to Assumption 1, we have:

$$
\begin{aligned}
&f(Y) - f(X) - tr(\nabla f(X)^\top (Y - X)) \\
=& \int_0^1 \nabla tr(f(Y + t(Y - X))^\top (Y - X))dt - tr(\nabla f(X)^\top (Y - X)) \\
\leq& ||\int_0^1 (\nabla f(Y + t(Y - X)) - \nabla f(X))dt|| \times ||Y - X|| \\
\leq& \int_0^1 L||t(Y - X)||dt \times ||Y - X|| \\
\leq& \frac{L}{2}||Y - X||^2
\end{aligned}
\tag{28}
$$

Then, we have

$$
\begin{aligned}
f(Z_{k+1}) \leq& f(Z_k) + tr(\nabla f(Z_k)^\top (Z_{k+1} - Z_k)) + \frac{L}{2}||Z_{k+1} - Z_k||^2 \\
=& f(Z_k) + tr(\nabla f(Z_k)^\top (Z_{k+1} - Z_k)) + \frac{L}{2}||\frac{\alpha}{1-\beta}\mathcal{G}_{\mathcal{M}}(X_k) - \frac{\alpha^2}{1-\beta}U||^2 \\
\leq& f(Z_k) + tr(\nabla f(Z_k)^\top (Z_{k+1} - Z_k)) + \frac{L\alpha^2}{(1-\beta)^2}||\mathcal{G}_{\mathcal{M}}(X_k)||^2 + \frac{L\alpha^4}{(1-\beta)^2}||U||^2 \\
\leq& f(Z_k) - \frac{\alpha}{1-\beta}tr(\mathcal{G}_{\mathcal{M}}(X_k)^\top \nabla f(Z_k)) + \frac{\alpha^2}{1-\beta}tr(U^\top \nabla f(Z_k)) + \frac{L\alpha^2}{(1-\beta)^2}G^2 \\
&+ \frac{L\alpha^4}{(1-\beta)^2}D^2 \\
\leq& f(Z_k) - \frac{\alpha}{1-\beta}tr(\mathcal{G}_{\mathcal{M}}(X_k)^\top \nabla f(Z_k)) + \frac{\alpha^2}{2(1-\beta)}(||U||^2 + ||\nabla f(Z_k)||^2) + \frac{L\alpha^2}{(1-\beta)^2}G^2 \\
&+ \frac{L\alpha^4}{(1-\beta)^2}D^2 \\
\leq& f(Z_k) - \frac{\alpha}{1-\beta}tr(\mathcal{G}_{\mathcal{M}}(X_k)^\top \nabla f(Z_k)) + \frac{\alpha^2}{2(1-\beta)}(D^2 + G^2) \\
&+ \frac{L\alpha^2}{(1-\beta)^2}G^2 + \frac{L\alpha^4}{(1-\beta)^2}D^2
\end{aligned}
\tag{29}
$$

By taking expectation over the both sides, we have:

$$\mathbb{E}[f(Z_{k+1}) - f(Z_k))]$$

$$\leq \mathbb{E}[-\frac{\alpha}{1-\beta}tr(\nabla f(Z_k)^\top \nabla_\mathcal{M} f(X_k))] + \frac{\alpha^2}{2(1-\beta)}(D^2 + G^2) + \frac{L\alpha^2}{(1-\beta)^2}G^2 + \frac{L\alpha^4}{(1-\beta)^2}D^2$$

$$\leq \mathbb{E}[-\frac{\alpha}{1-\beta}tr(\nabla f(Z_k) - \nabla f(X_k))^\top \nabla_\mathcal{M} f(X_k) - \frac{\alpha}{1-\beta}tr(\nabla f(X_k)^\top \nabla_\mathcal{M} f(X_k))]$$

$$+ \frac{\alpha^2}{2(1-\beta)}(D^2 + G^2) + \frac{L\alpha^2}{(1-\beta)^2}G^2 + \frac{L\alpha^4}{(1-\beta)^2}D^2$$

$$= \mathbb{E}[-\frac{\alpha}{1-\beta}tr(\nabla f(Z_k) - \nabla f(X_k))^\top \nabla_\mathcal{M} f(X_k) - \frac{\alpha}{1-\beta}||\nabla_\mathcal{M} f(X_k)||^2]$$

$$+ \frac{\alpha^2}{2(1-\beta)}(D^2 + G^2) + \frac{L\alpha^2}{(1-\beta)^2}G^2 + \frac{L\alpha^4}{(1-\beta)^2}D^2 \tag{30}$$

By noticing that:

$$- \frac{\alpha}{1-\beta}(\nabla f(Z_k) - \nabla f(X_k))^\top \nabla_\mathcal{M} f(X_k)$$

$$\leq \frac{1}{2L}||\nabla f(Z_k) - \nabla f(X_k)||^2 + \frac{L\alpha^2}{2(1-\beta)^2}||\nabla_\mathcal{M} f(X_k)||^2 \tag{31}$$

Then

$$\mathbb{E}[f(Z_{k+1}) - f(Z_k))]$$

$$\leq \frac{1}{2L}\mathbb{E}||\nabla f(Z_k) - \nabla f(X_k)||^2 + (\frac{L\alpha^2}{2(1-\beta)^2} - \frac{\alpha}{1-\beta})\mathbb{E}||\nabla_\mathcal{M} f(X_k)||^2$$

$$+ \frac{\alpha^2}{2(1-\beta)}(D^2 + G^2) + \frac{L\alpha^2}{(1-\beta)^2}G^2 + \frac{L\alpha^4}{(1-\beta)^2}D^2 \tag{32}$$

According to the Lipschitz continuous property in Assumption 1, we have:

$$||\nabla f(Z_k) - \nabla f(X_k)||^2 \leq L^2||Z_k - X_k||^2$$

$$= \frac{L^2\beta^2}{(1-\beta)^2}||X_k - X_{k-1}||^2$$

$$= \frac{L^2\beta^2}{(1-\beta)^2}||\alpha M_k + \frac{1}{2}\alpha^2 \frac{d^2 Y(\gamma_k \alpha)}{d\alpha^2}||^2$$

$$\leq \frac{2L^2\beta^2}{(1-\beta)^2}(||\alpha M_k||^2 + ||\frac{1}{2}\alpha^2 \frac{d^2 Y(\gamma_k \alpha)}{d\alpha^2}||^2)$$

$$\leq \frac{2L^2\alpha^2\beta^2}{(1-\beta)^2}(\frac{G^2}{(1-\beta)^2} + \frac{\alpha^2 C^2}{4}) \tag{33}$$

Therefore,

$$\mathbb{E}[f(Z_{k+1}) - f(Z_k))]$$

$$\leq -B\mathbb{E}||\nabla_\mathcal{M} f(X_k)||^2 + \alpha^2 B' \tag{34}$$

where

$$B = \frac{\alpha}{1-\beta} - \frac{L\alpha^2}{2(1-\beta)^2} \tag{35}$$

$$B' = \frac{L\beta^2}{(1-\beta)^2}(\frac{G^2}{(1-\beta)^2} + \frac{\alpha^2 C^2}{4}) + \frac{D^2 + G^2}{2(1-\beta)} + \frac{LG^2}{(1-\beta)^2} + \frac{L\alpha^2}{(1-\beta)^2}D^2 \tag{36}$$

Since $\alpha \leq \frac{1-\beta}{L}$, then

$$
\begin{aligned}
B &= \frac{\alpha}{1-\beta} - \frac{L\alpha^2}{2(1-\beta)^2} \\
&= \frac{\alpha}{1-\beta}\left(1 - \frac{\alpha L}{2(1-\beta)}\right) \\
&\geq \frac{\alpha}{2(1-\beta)} > 0
\end{aligned}
\tag{37}
$$

By summing Eq.(34) over all $k$, we have

$$
B\sum_{k=0}^{t} \mathbb{E}||\nabla_\mathcal{M} f(X_k)||^2 \leq \mathbb{E}[f(Z_0) - f(Z_{t+1})] + (t+1)\alpha^2 B'
$$

$$
\leq f(Z_0) - f_* + (t+1)\alpha^2 B'
\tag{38}
$$

Then

$$
\min_{k=0,\cdots,t} \mathbb{E}[||\nabla_\mathcal{M} f(X_k)||^2] \leq \frac{f(Z_0) - f_*}{(t+1)B} + \alpha^2\frac{B'}{B}
\tag{39}
$$

$$
\min_{k=0,\cdots,t} \mathbb{E}[||\nabla_\mathcal{M} f(X_k)||^2] \leq \frac{2(f(Z_0) - f_*)(1-\beta)}{(t+1)\alpha} + \alpha 2B'(1-\beta)
\tag{40}
$$

Use the fact that $\alpha = \min\{\frac{1-\beta}{L}, \frac{A}{\sqrt{t+1}}\}$, and notice that $Z_0 = X_0$, therefore,

$$
\begin{aligned}
&\min_{k=0,\cdots,t} \mathbb{E}[||\nabla_\mathcal{M} f(x_k)||^2] \\
&\leq \frac{2(f(X_0) - f_*)(1-\beta)}{t+1}\max\left\{\frac{L}{1-\beta}, \frac{\sqrt{t+1}}{A}\right\} + \frac{2AB'(1-\beta)}{\sqrt{t+1}} \\
&\leq \frac{2(f(X_0) - f_*)(1-\beta)}{t+1}\max\left\{\frac{L}{1-\beta}, \frac{\sqrt{t+1}}{A}\right\} \\
&\quad + \frac{2A(1-\beta)}{\sqrt{t+1}}\left[\frac{L\beta^2}{(1-\beta)^2}\left(\frac{G^2}{(1-\beta)^2} + \frac{A^2 C^2}{4(t+1)}\right)\right. \\
&\quad \left. + \frac{D^2 + G^2}{2(1-\beta)} + \frac{LG^2}{(1-\beta)^2} + \frac{LA^2}{(t+1)(1-\beta)^2}D^2\right]
\end{aligned}
\tag{41}
$$

Therefore, $\min_{k=0,\cdots,t} \mathbb{E}[||\nabla_\mathcal{M} f(X_k)||^2] = o(\frac{1}{\sqrt{t+1}}) \to 0$, as $t \to \infty$.

$\square$

