# OpenReview forum: "Efficient Riemannian Optimization on the Stiefel Manifold via the Cayley Transform"
_ICLR.cc/2020/Conference — Accept (Poster)_

### Official Review · AnonReviewer3 · 2019-10-23
**Official Blind Review #3**

**Rating:** 6

**Review:**

Summary:

This paper proposes an efficient method to perform Riemannian optimization over the Stiefel manifold using an efficient computation of the Cayley transform via fixed point iteration. While simple, the method is to my knowledge novel. Empirically, the authors validate that their method is able to tightly enforce the orthogonality constraint with a reasonable computational budget. Further experiments highlight the benefits of orthogonality in deep learning but the explanations feels lacking. The paper is missing references to some related work but is otherwise well written.

Overall:

1) In abstract, "This amounts to Riemannian optimization on the Stiefel manifold". Other approaches exist to enforce orthogonality throughout the network. In general, I felt that this paper was missing related work which enforces orthogonality constraints in deep learning. Some examples: [1,2,3,4] though this is by no means exhaustive.

2) In order for the fixed-point iteration to have guaranteed convergence, the "learning rate" for the contraction mapping depends on the largest singular value of the weight matrix $W_k$. Could you please clarify how this is computed in practice?

3) I am not convinced that the second change made to the Adam algorithm is reasonable (at least, not if we wish to continue calling the algorithm Adam). Adam preconditions the gradient by an estimate of the diagonal of the Fisher information matrix. Algorithm 2 presented is closer to a spherical approximation to the Fisher. The algorithm still looks sensible to me but the name is perhaps inaccurate.


4) I am a little unsure of the motivation behind the experiments. For experiments validating the efficiency of the proposed method this is clear but for the classification experiments this is less obvious. Are the authors claiming that orthogonality is a good regularizer? Or is the only benefit in optimization? I did not understand the proposed explanation that that the orthogonal weights do not affect eachother during backpropagation --- could you please clarify? Even if this were true, why would this encourage exploration? There is existing literature which suggests that orthogonal networks will be easier to train (see e.g. [5] and others).

5) The orthogonality of the convolutional layers is enforced by reshaping the kernel and imposing an orthogonality constraint there instead. Unfortunately, this does not guarantee that the actual convolution operator is orthogonal (see e.g. [2] for an example in 1D and [5] which characterizes orthogonal convolutions correctly).

Minor:

- Section 2, TYPO: "non-suquare parameter matrices".
- The momentum $M_t$ is used below Equation 2 before it is defined.

References:

[1] Cheap Orthogonal Constraints in Neural Networks: A Simple Parametrization of the Orthogonal and Unitary Group, Mario Lezcano-Casado and David Martínez-Rubio
[2] Parseval Networks: Improving Robustness to Adversarial Examples, Moustapha Cisse, Piotr Bojanowski, Edouard Grave, Yann Dauphin, and Nicolas Usunier
[3] Orthogonal Recurrent Neural Networks with Scaled Cayley Transform, Kyle Helfrich, Devin Willmott, and Qiang Ye
[4] Variational Inference with Orthogonal Normalizing Flows, Leonard Hasenclever, Jakub M. Tomczak, Rianne van den Berg, and Max Welling
[5] Dynamical Isometry and a Mean Field Theory of CNNs: How to Train 10,000-Layer Vanilla Convolutional Neural Networks, Lechao Xiao, Yasaman Bahri, Jascha Sohl-Dickstein, Samuel S. Schoenholz,  and Jeffrey Pennington

**Experience Assessment:**

I have read many papers in this area.

**Review Assessment: Checking Correctness Of Derivations And Theory:**

I assessed the sensibility of the derivations and theory.

**Review Assessment: Checking Correctness Of Experiments:**

I carefully checked the experiments.

**Review Assessment: Thoroughness In Paper Reading:**

I read the paper at least twice and used my best judgement in assessing the paper.

---

> ### Author Response · Authors · 2019-11-15
> **Thanks for your constructive feedback. Below, we address your detailed questions.**
>
>
> Q1:” ...this paper was missing related work which enforces orthogonality constraints in deep learning. Some examples: [1,2,3,4]”.
>
> A1: Thank you for the heads up. We have added them to the related work.
>
>
> Q2:”the ‘learning rate’ for the contraction mapping depends on the largest singular value of the weight matrix. Could you please clarify how this is computed in practice ”
>
> A2: In Theorem 1, we show that as long as \alpha \in (0, \min \{1, \frac{1}{||w||}\}), the contraction mapping will converge. So we do not need to compute the largest singular value of the weight matrix in practice.
>
>
> Q3:”Algorithm 2 presented is closer to a spherical approximation to the Fisher. Adam still looks sensible to me but the name is perhaps inaccurate”
>
> A3:  In section 4.2, the paper clarifies that “we use a manifold-wise adaptive learning rate”, which addresses your comment about “spherical approximation to the Fisher”.
>
>
> Q4: I am a little unsure of the motivation behind the experiments. Are the authors claiming that orthogonality is a good regularizer? Or is the only benefit in optimization?
>
> A4: In introduction and related work, it shows that benefits of orthogonality in improving both classification accuracy and training convergence rate are reported in Bansal et al. (2018), Huang et al. (2018a), Cogswell et al. (2015), Arjovsky et al. (2016) and Zhou et al. (2006).  For the second question if orthogonality only benefit in optimization, this is beyond our paper scope, i.e. the efficiency of optimization on the Stiefel manifold. We will leave the second question for future research.
>
>
> Q5: “The orthogonality of the convolutional layers is enforced by reshaping the kernel and imposing an orthogonality constraint there instead. Unfortunately, this does not guarantee that the actual convolution operator is orthogonal ”
>
> A5: Imposing an orthogonality constraint to the reshaped kernel orthogonalizes the kernel of different out channels. And it also keep the norm of the original feature map nearly unchanged. This is because convolution can be performed by matrix multiplication of a reshaped feature map and a reshaped kernel. During the feature map reshaping process, all pixels in the original feature map appear the same times in the reshaped feature map except the boundary pixels may appear fewer times. Therefore, imposing an orthogonality constraint on the reshaped kernel is reasonable.
>
> Besides, the prior work (Bansal et al.,2018, Huang et al., 2018a, Huang et al., 2018b) also uses the same setting. We follow them as a fair comparison.

---

> > ### Comment · AnonReviewer3 · 2019-11-15
> > **Response**
> >
> > My response here is a little hasty to allow time for further response from the authors.
> >
> > On Q2: My understanding on first reading was that the formula for alpha depended on the largest singular value of W (in particular, it cannot exceed $1/||W||_2$). However, it seems that the convergence results are actually given in terms of the Frobenius norm (whichever norm it is, this should be stated explicitly). In this case, the choice of alpha instead requires computing the Frobenius norm of $W$.
> >
> > On Q3: My concern was not that the paper  is not clear on the actual algorithm used. I was pointing out that the algorithm presented is not really "Cayley ADAM", as ADAM necessarily has an adaptive learning rate per parameter. This is not a major issue, but could be slightly misleading.
> >
> > On Q5: I would argue that in fact this reshaping can impose a significant constraint. As you describe, the convolutional operator can be represented as a linear map depending on the kernel matrix _and_ the padding and stride. This is important, and there is existing work which explores this limitation. [1] gives a more careful treatment of this problem. [2] explores the 1D case and points out that the bound is only tight up to a constant. [3] discusses many alternatives to imposing Lipschitz constraints on convolutions, including the reshaped kernel method.
> >
> > I am not arguing that this does not encourage a Lipschitz bound, but the convolution itself will not be orthogonal in general.
> >
> > [1] "The Singular Values of Convolutional Layers" Hanie Sedghi, Vineet Gupta, and Philip M. Long
> > [2] "Parseval Networks: Improving Robustness to Adversarial Examples", Moustapha Cisse, Piotr Bojanowski, Edouard Grave, Yann Dauphin, and Nicolas Usunier
> > [3] "Preventing Gradient Attenuation in Lipschitz Constrained Convolutional Networks" Qiyang Li, Saminul Haque, Cem Anil, James Lucas, Roger Grosse, Jörn-Henrik Jacobsen

---

> > > ### Author Response · Authors · 2019-11-15
> > > **Thanks for your comment. Below, we address your detailed questions.**
> > >
> > >
> > > Q1: "the convolution itself will not be orthogonal in general."
> > >
> > > A1: We admit it might change the norm of feature maps, but it keeps the kernels of different output channels to be orthogonal to each other. There is only a small scaling factor difference with the strict orthogonal map. Thanks for the heads up. We will leave the efficiency of the method in [1,2] for our future research.

---

### Official Review · AnonReviewer2 · 2019-10-23
**Official Blind Review #2**

**Rating:** 3

**Review:**

Summary
This paper aims to improve upon current solutions for optimizing neural networks with orthonormal convolutional kernels/MLP layers. Optimizing neural networks while restricting the parameter matrices to remain orthonormal/on the Stiefel manifold is said to lead to faster convergence in terms of the number of epochs, and could reduce overfitting. One way to enforce orthonormality is to use the Cayley transform, which requires a matrix inversion that becomes expensive for large weight matrices in neural networks. The authors instead propose an iterative approximation to the Cayley transform that does not require a matrix inversion, and empirically only requires two iteration steps to lead to similar precision as a closed-form Cayley transform with numerical inverse. The authors combine this with SGD + momentum and ADAM by first performing the update step in Euclidean space and afterwards projecting the result back onto the stiefel manifold by using the iterative Cayley transform. They provide one assumption and two theorems on the convergence of the iterative approximation and its effect on optimization. The method is evaluated on classification for cifar10 and cifar100 and for modeling the hidden-to-hidden transition matrix in an RNN trained on MNIST.

Decision:
Weak reject. Although the motivation of the paper is sound. The empirical validation of the proposed method is insufficient. For instance, assumptions that are the basis of one of the theorems are violated in the experiments and converges are only shown as a function of epoch and not wall clock time.

Supporting arguments for decision:
The main issue with the paper is that the claims made are not sufficiently supported. I have the following three main issues with the evaluation:
1)  Assumption 1, which is required to prove convergence of the proposed Cayley SGD/ADAM optimizer, appears to be violated in experiments. The assumption states that the gradient of the objective function is Lipschitz continuous. However, in the VGG and wide Resnets ReLU’s are used. The derivative of a ReLU is a step function, which is not Lipschitz continuous. Therefore the objective function used in the experimental validation violates assumption 1. Now, I can imagine that perhaps in practice this does not matter too much, but surely the evaluation is not correct according to the theoretical claim. The authors should clarify this and experiments should be done with other activation functions that do not violate the assumption.

2) The paper claims to improve the convergence speed, as compared to baseline Euclidian SGD+momentum and ADAM, but learning curves are only shown as a function of epoch, and not of wall clock time. Table 3 displays per-epoch training times and clearly shows that when compared to SGD+momentum and ADAM the runtime is slowed down by a factor of 2. I am not convinced that if figure 1 would be plotted as a function of wall clock time, the proposed method would actually come out as having converged faster.

3) The learning rate decay schedule used in obtaining the results in figure 1 and table 1 seems more optimized for the proposed methods than for the baseline SGD and ADAM optimizers. ADAM and SGD have reached a plateau “earlier” (in terms of epoch number), and could have benefited from a decay in learning rate earlier on. It is also extremely hard to see what is going on after the 50th epoch due the scale of the plot in figure 1. Please zoom in on epoch 50 and onwards or also show a plot in log scale. The total number of epochs is also the same for all methods. Perhaps it would also be more fair to give every method the same total budget of wall clock time.

The following issues are more minor, but I would still like to see them addressed.
1) In the last section of the related work, the work by Wen & Yin is said to be “not suitable for training common deep neural networks”. This gives the impression that this method simply cannot be used for VGG/wide resnets. However, table 3 certainly shows results for the work by Wen & Yin for a large wide resnet with a per epoch run time that is much better than the Cayley closed form w/o momentum baseline.
2) In table 3, why is the SO training time per epoch slower than the Cayley SGD/ADAM optimization?

---- Post Rebuttal Update ----

Assumption one in the paper states that the derivative of the objective function must be Lipschitz continuous - the authors use ReLU in all experiments whose derivative is not Lipschitz continuous.

The authors address this concern with the following (page 6 in paper)

"... For some models using ReLU, the derivative of ReLU is Lipschitz continuous almost everywhere with an appropriate Lipschitz constant in Assumption 1 , except for a small neighborhood around 0, whose measure tends to 0. Such cases do not affect either analysis in theory or training in practice."

On the theory side: If the authors want to rely on the assumption that for a measure zero part of the domain the function is not Lipschitz, then the authors should adjust assumption 1 to loss functions whose derivatives are locally Lipschitz continuous, and this will require a redo of the proofs under this new assumption.
From the experimental side: they provide no direct comparison with models where assumption 1 does hold, e.g by comparing ReLU with a scaled SoftPlus as suggested in the review. So the claim that it does not matter in practice is not completely supported.

As this is an optimization paper the wall-clock convergence rates are important and was not included in the initial submission. The authors have addressed this by adding figure 2 in the appendix, which shows convergence rates from 0 - 20000 seconds. Presumably this is a subset (likely epoch 0 to 80) of the data shown in figure 1 showing 0 to 200 epochs. Why they choose not to show the full data is not completely clear to me. The authors claim that Cayley Adam catches up with Adam after 12000 seconds which might be true for cifar10 (not really visible in the figure) but is definitely not true for cifar100 where Adam seems to be better all the time.

In summary: Claims that deviating from assumption 1 does not impact theory or experiment are not supported. Important run time experiments are relegated to the appendix and only  incomplete training curves as a function of time are shown, which is potentially misleading.


**Experience Assessment:**

I have read many papers in this area.

**Review Assessment: Checking Correctness Of Derivations And Theory:**

I carefully checked the derivations and theory.

**Review Assessment: Checking Correctness Of Experiments:**

I carefully checked the experiments.

**Review Assessment: Thoroughness In Paper Reading:**

I read the paper thoroughly.

---

> ### Author Response · Authors · 2019-11-15
> **We appreciate your constructive feedback. We have clarified Lipschitz continuous of ReLU, zoomed the loss curve, and added a figure of runtimes per your suggestions. Below, we address your detailed questions.**
>
>
> Q1:”The assumption states that the gradient of the objective function is Lipschitz continuous,..., The derivative of a ReLU is a step function, which is not Lipschitz continuous. ”
>
> A1: By choosing an appropriate Lipschitz constant L, the derivative of a ReLU is Lipschitz continuous almost everywhere, except for a small neighborhood around 0, i.e. (-\epsilon, \epsilon). In practice, this does not affect training.  We have revised the paper to incorporate this discussion (beginning of page 6).
>
>
> Q2:”The paper claims to improve the convergence speed, ..., but learning curves are only shown as a function of epoch, and not of wall clock time. ”
>
> A2: Thank you for the feedback. We have added a new figure of runtimes in the appendix. The figure shows that our approach is the fastest among methods that also address orthonormality. Although SGD and ADAM are faster, as they do not pay the price of enforcing orthogonality constraints,  Table 3 shows that classification accuracies of SGD and ADAM on test data are inferior to ours.
>
>
> Q3: “Please zoom in on epoch 50 and onwards or also show a plot in log scale.”
>
> A3: We have revised the paper by replacing Figure 1 with a zoomed loss curve for epoch 40-100.
>
>
> Q4: “Wen & Yin is said to be ‘not suitable for training common deep neural networks’. However, table 3 certainly shows results for the work by Wen & Yin”
>
> A4: Thank you for pointing this out. The strong statement about Wen & Yin has been removed in our revision. The original statement was to emphasize the inefficiency of Wen & Yin. Table 3 shows that our approach outperforms Wen & Yin in terms of both efficiency and accuracy, which corresponds to the original statement.
>
>
> Q5: “In table 3, why is the SO training time per epoch slower than the Cayley SGD/ADAM optimization”
>
> A5: Bansal et al., 2018 compute the SO regularization in both the forward and backward passes, while our methods only add additional computations in the backward pass. For Table 3, we used the public implementation of Bansal et al., 2018.

---

> > ### Comment · AnonReviewer2 · 2019-11-15
> > **Response**
> >
> > Thank you for your extensive rebuttal and revision of the paper.
> >
> > Thank you for the plots of convergence as a function of time. In my opinion these plots should be in the main paper, and not in the appendix. You are targeting faster convergence so placing the results as a function of time in the appendix makes them seem inferior, even though I would argue that it is actually more important than the convergence as a function of epoch.
> >
> > The argument that the vanilla Adam and SGD+momentum underperform in terms of classification might still be because you give every method the same budget in terms of epochs and not in terms of wall clock time.
> >
> > Thank you for adding zoomed plots in figure 1, but why are the zoomed plots in figure 1 now only showing results up to epoch 100 and not up to 200 as previously?. This now makes the difference in results between vanilla ADAM and Cayley Adam seem much more pronounced that it was in the original figure that included results with more epochs.
> >
> > I appreciate the included discussion of ReLU’s. I’m aware that there is only a small region where the derivative of the ReLU is not Lipschitz continuous. However, as your convergence results completely rely on the Lipschitz continuity assumption, experiments with another activation function that does not have this problem would make the paper more convincing, as it would bridge the gap between theory and experiment. To show that in practice this problem with ReLU’s does not harm training, one could for instance perform an experiment where the ReLU’s are replaced with a softplus that has an additional parameter that determines how close you are to a ReLU.

---

> > > ### Author Response · Authors · 2019-11-15
> > > **Thanks for your comment. Below, we address your detailed questions.**
> > >
> > >
> > > Q1: "the plots of convergence as a function of time...  should be in the main paper, and not in the appendix. You are targeting faster convergence "
> > >
> > > A1: The main goal of our paper is to show efficient optimization of orthonormal optimization on the Stiefel manifold, not to compete for the running time of baselines SGD and Adam.  Also, the comparison of convergence in terms of epoch will give an insight into the difference of orthonormal methods and non-orthonormal methods. So, we think Figure 1 for learning curve in terms of epoch is also important in our paper.
> > >
> > >
> > > Q2: "The argument that the vanilla Adam and SGD+momentum underperform in terms of classification might still be because you give every method the same budget in terms of epochs and not in terms of wall clock time. "
> > >
> > > A2: I am afraid this is not true. Firstly, as shown in Figure 1, the vanilla Adam and SGD+momentum already converge at the 200 epoch.  Furthermore, Figure 2 shows Cayley SGD and Cayley Adam catch up with the vanilla Adam and SGD after 12000 seconds.  Therefore, simply giving the vanilla Adam and SGD+momentum more time for training will not improve their performance.
> > >
> > >
> > > Q3: "but why are the zoomed plots in figure 1 now only showing results up to epoch 100 and not up to 200 as previously?."
> > >
> > > A3: We thought it might be more clear to show the learning curve from epoch 40 to epoch 100. Now, we replace figure 1 with zoomed plots from epoch 40 to 200.
> > >
> > >
> > > Q4: " as your convergence results completely rely on the Lipschitz continuity assumption, ..., it would bridge the gap between theory and experiment"
> > >
> > > A4: As shown in the begining of page 6, with an appropriate Lipschitz constant L in Assumption 1, the measure of the small neighbourhood that is not Lipschitz continous tends to 0. Therefore, it does not affect either analysis in theory or training in practice.

---

### Official Review · AnonReviewer1 · 2019-10-26
**Official Blind Review #1**

**Rating:** 6

**Review:**

The paper proposes a fast algorithm to train a NN under orthogonality
constraints on the weights for each layer. Using a new retraction
the paper proposes an adaptation of SGD or ADAM on the Stiefel
manifold.

The idea of the paper is to use a truncated fixed point iteration to
obtain the Cayley transform. By doing so one has an approximation
of a cheap retraction by just doing some matrix vector products
(no matrix inversion or SVD needed). The idea is seducing but I
see some difficulties due to this approximation.

Theorem 2 proves that the relative gradient tends to zero but
this does not control how "orthogonal" are the obtained weights.
Basically I fear that the proposed iterative solver moves
the iterates away from the manifold although some empirical
results in Table 5 suggest otherwise.

Figure 1 should be replaced or complemented by a convergence plot
in running time. What matters is that test error decreases
faster as function of time (not epoch). Because of this,
experiments are not fully convincing.

typos:

- non-suquare -> non-square

**Experience Assessment:**

I have read many papers in this area.

**Review Assessment: Checking Correctness Of Derivations And Theory:**

I assessed the sensibility of the derivations and theory.

**Review Assessment: Checking Correctness Of Experiments:**

I assessed the sensibility of the experiments.

**Review Assessment: Thoroughness In Paper Reading:**

I read the paper at least twice and used my best judgement in assessing the paper.

---

> ### Author Response · Authors · 2019-11-15
> **We appreciate your constructive feedback. We have fixed the typos, and added a figure of runtimes per your suggestions. Below, we address your detailed questions.**
>
>
> Q1:”I fear that the proposed iterative solver moves the iterates away from the manifold”
>
> A1: In Theorem 1, we show that our approach theoretically achieves orthonormality faster than other approximation algorithms including the Newton iterative and Neumann Series. In Table 5, we also empirically show that our approach achieves good orthonormality in terms of numerical precision. Furthermore, for every 1000 iterations in our implementation, we use the QR decomposition that orthogonalizes the parameter matrix for removing the potential rounding error. This implementation step is included in our estimation of runtime.
>
> Q2:”Figure 1 should be replaced or complemented by a convergence plot in running time”
>
> A2: Thank you for the feedback. We have added a new figure of runtimes in the appendix. The figure shows that our approach is the fastest among methods that also address orthonormality. Although SGD and ADAM are faster, as they do not pay the price of enforcing orthogonality constraints,  Table 3 shows that classification accuracies of SGD and ADAM on test data are inferior to ours.

---

### Public Comment · ~Mario_Lezcano-Casado1 · 2019-10-22
**Interesting paper. Lacks a more in-depth literature review and comparisons with previous methods**

First of all, thank the authors for this very interesting paper. I think that the idea is nice, but I am worried about a few points in it.

At first sight, I do not see what is the role of the Cayley transform in the algorithm. It looks like this method could be posed in more generality for any retraction on the Stiefel manifold. I would also be interested in knowing whether the ideas in the proofs could be extended to this setting and, more generally, any Riemannian manifold together with a retraction.

Probably the two main problems I see in this paper are:

1) A strong discussion at the beginning for the need of this approximation to the Cayley transform. in section 4.3 in [1], it is shown that one can even use the exponential parametrisation in the context of deep learning without any noticeable computational penalty (it runs just as fast as any other retraction) when implementing it correctly. For this reason, I fail to see the need of an approximation of something that can be computed exactly. I understand that more efficient methods are appealing in other optimisation problems, or when they are used in, for example, alternating optimisation, but the applications in this paper are mostly concerned with neural networks.

2) The deep learning community has been looking rather closely at methods to optimise on manifolds in the context of deep learning. Most of the recent ones are missing both in the experiments section and the literature review. See for example [1,2,3,4] among others. In particular, it does not compare against [2], when this paper exactly implements a trivialisation using the Cayley transform, which has some similarities with this method. Also in the experiments section it mentions comparisons with state of the art algorithms, but in the sequential MNIST  it compares against Wisdom, achieving a 96.9 accuracy, while in [1] the method achieves an accuracy of 98.7, and in [3] it achieves one of 99.1.

In summary, I think that the paper points in the right direction, but it is still missing some fundamental work namely, a more general framework, a stronger motivation, and a more in-depth literature review and comparison in the experiments section.

I would love to see a paper like this, on which the ideas are stated for general manifolds together with an arbitrary retraction, where the "vector transport" is formalised as the induced parallel transport by the pullback metric.

[1]: Cheap Orthogonal Constraints in Neural Networks: A Simple Parametrization of the Orthogonal and Unitary Group (2019)
[2]: Orthogonal Recurrent Neural Networks with Scaled Cayley Transform (2017)
[3]: Trivializations for Gradient-Based Optimization on Manifolds (2019)
[4]: Non-normal Recurrent Neural Network (nnRNN): learning long time dependencies while improving expressivity with transient dynamics (2019)

---

> ### Author Response · Authors · 2019-10-28
> **Thanks for the comments. However, there seems to be some general misunderstanding about our scope and contributions.**
>
> We appreciate the reviewer’s comments. There seems to be some general misunderstanding about our scope and contributions, and we would like to clarify them first. Then, below, we address more detailed comments point by point.
>
> The reviewer cites a number of approaches to Riemannian optimization on Orthogonal and Unitary Group [1]-[4] that we have not cited in the paper. We find that [1]-[4] are not closely related to our approach. Unlike ours, these approaches are aimed at optimizing *square* parameter matrices. Thus,  their application is restricted to optimizing parameters of only special deep architectures (e.g., transition matrix of RNNs), but cannot be easily extended to common CNNs with rectangular parameter matrices where input and output dimensions are different. In contrast, we address Riemannian optimization on the Stiefel manifold for *non-square* orthonormal matrices, as explained in the third paragraph in Introduction. In comparison with [1]-[4], our approach is suitable for a wider range of common deep learning models, e.g., for both CNNs and RNNs.
>
> Q1:  “in section 4.3 in [1], it is shown that one can even use the exponential parameterization in the context of deep learning without any noticeable computational penalty (it runs just as fast as any other retraction) when implementing it correctly. For this reason, I fail to see the need of an approximation of something that can be computed exactly.”
>
> A1:  Our better efficiency and better numerical precision in practice are immediate two reasons that the reviewer fails to see, among others described in the paper. Regarding efficiency, it is straightforward to see that the exponential parameterization in [1] involves a greater number of matrix operations than the closed-form Cayley transform. Specifically, this parameterization involves computing a matrix inverse, and as such is quite inappropriate for deep learning over many iterations. In our paper, Table 3 shows that our approach is more computationally efficient than the closed-form Cayley transform, and other orthonormal methods for CNNs, without compromising performance. Furthermore, unlike the closed-form Cayley transform, our method has no numerical precision issues. Table 5 shows that we achieve a better numerical precision than the closed-form Cayley transform in practice.
>
> Q2: “Most of the recent ones are missing both in the experiments section and the literature review. See for example [1,2,3,4] among others. In particular, it does not compare against [2], when this paper exactly implements a trivialisation using the Cayley transform, which has some similarities with this method.”
>
> A2: We find that  [1,2,4] are not closely related, and therefore are not mentioned in the paper. In particular, [1,2] do not use the closed form Cayley transform. [4] does not address an orthogonal optimization of RNNs, i.e., [4] addresses a different problem from ours. The main purpose of our experiments with RNNs experiments is to show that our approach achieves higher computational efficiency for Unitary RNNs without compromising accuracy in comparison to the baselines which also use the closed-form Cayley transform.  Finally, [3] is a very recent paper that we could have not compared with before the ICLR deadline. We can easily mention [1,2,3,4] in the literature review.
>
>
>
> Q3: “the paper ... is still missing some fundamental work namely, a more general framework”
>
> A3:  We appreciate the feedback, but our paper should be viewed as an important step toward a more general framework. Our contribution is an efficient formulation of Riemannian optimization using the Cayley transform on the Stiefel manifold. A more general formulation to other retraction or parameterization types (let alone to an arbitrary retraction as the reviewer wishes) is a long-standing open problem.

---

> > ### Public Comment · ~Mario_Lezcano-Casado1 · 2019-11-03
> > **All the papers cited above, either apply directly to the Stiefel manifold ([3]) or can be trivially adapted to the Stiefel manifold ([1,2,4])**
> >
> > >We find that [1-4] are not closely related to our approach. Unlike ours, these approaches are aimed at optimizing *square* parameter matrices.
> >
> > The reference [3] does address exactly the problem in the Stiefel manifold, together with any other provided that one has access to a retraction. The exact computations are even carried in Section E.3.1. On the other hand, I do agree that this reference is rather new and I understand that the authors wouldn't have had time to add it.
> >
> > > [1,2] are not closely related to our approach
> > These references perform optimization over SO(n) in exactly the same context and using the same experiments as [5,6], references that are cited in the related work section and are compared against in the experiments.
> >
> > Note that in [6], what the authors refer to as the Stiefel manifold, is actually U(n) (cf. eq 5). In [5], the authors, what the authors refer to as the Stiefel manifold is SO(n), as they work with skew-symmetric matrices (cf. eq 9).
> >
> > References [1,2] largely improve on [5,6] on every task MNIST task, so not comparing against them does not seem fair, given that they yield much better results than those presented in this paper.
> >
> > On a separate note, saying that optimization on SO(n) and St(n,k) are "not closely related" might also be a bit of a stretch given that, if one has a method to optimize over SO(n), one can get a method to optimize over St(n,k) just looking at the first k columns of the matrix in SO(n) being optimized. This being said, it would have been reasonable to include also a comparison against the methods presented in [1,2] using this rather natural approach, given that the methods in [1,2] conform the state of the art for optimization over SO(n) for deep learning.
> >
> > > Complexity gains
> > One way of exponential and the Cayley transform do not involve computing an inverse, but solving a well-conditioned linear system, which is rather computationally stable. But, as described in section 4.2 in [1], this is just one way of approximating the exponential of matrices (or the Cayley or any analytic function, for this matter), as one can use a Taylor approximant, and the analysis in this paper is independent of this choice.
> >
> > In the case of the exponential of a matrix of 64 bits, a Taylor approximant of degree 18 is enough to achieve machine-precision, which can be evaluated using 5 matrix multiplications (see [7]) if one finds the Padé approximant too slow. Another analysis can also be found in [8].
> >
> > On the other hand, as Section 4.3 in [1] points out, these finer approximations are unlikely to be necessary in the deep learning context, as the computation complexity of the forward-backward pass tends to be far more expensive than the computation of an approximant of a function for just one matrix.
> >
> >
> > [5] On orthogonality and learning recurrent networks with long term dependencies. (2017)
> > [6] Full-capacity unitary recurrent neural networks. (2016)
> > [7] An Arbitrary Precision Scaling and Squaring Algorithm for the Matrix Exponential (2018)
> > [8] Computing the matrix exponential with an optimized taylor polynomial approximation (2018)

---

> > > ### Author Response · Authors · 2019-11-15
> > > **Thanks for your comments. But there are no previous approaches that apply [1,2,4] to the Stiefel manifold, and they are beyond our scope, i.e. the efficency for optimization on the Stiefel manifold.**
> > >
> > > Q1:”References [1,2] largely improve on [5,6] on every task MNIST task, so not comparing against them does not seem fair”
> > >
> > > A1: Our main point is to show that we have improved the efficiency of the vanilla Cayley transform for optimization on the Stiefel manifold in deep learning without compromising performance. The methods in [1,2] are not as efficient as ours. We have added the references in our literature review, and we will address the efficiency of [1,2] for our future research.
> > >
> > > Q2:”One way of exponential and the Cayley transform do not involve computing an inverse, …, but this is just one way of approximating the exponential of matrices, ..., as one can use a Taylor approximant”
> > >
> > > A2: We have shown in Theorem 1 that our method achieves a faster convergence rate than the Taylor approximants. During the approximation process, our method achieves a lower error bound o(\alpha ^ (2+i)) at iteration i, while the Taylor approximants achieve o(\alpha ^ (i)) error bound at iteration i.

---

### Public Comment · ~Alexander_Mathiasen2 · 2020-01-21
**Reshaping of kernel.**

In secion 6.1, is there a reason the kernel was reshaped to $ c_{in}    \times   (c_{out}*h*w)$ instead of  $(c_{in} * h)   \times   (c_{out}*w)$?

---

> ### Author Response · Authors · 2020-01-21
> **Because we want the output channels to be orthonormal.**
>
> Thanks for your question. We reshape the kernel to $c_{out} \times (c_{in}*h*w)$ because we want the output channels to be orthonormal. Other forms like $c_{in} \times (c_{out}*h*w)$, instead, orthonormally distributed input channels to the output. $(c_{in}*h)\times (c_{out}*w)$ is mixing channels and spatial dimensions of the kernel, so it's hard to interpret this reshaping.

---

> > ### Public Comment · ~Alexander_Mathiasen2 · 2020-01-25
> > **How much does that matter?**
> >
> > Thanks for the quick reply.
> >
> > I agree that it would be "harder to interpret" the other reshaping.
> > Do you know of any empirical evaluation that compares the two reshapings?
> >
> > The reason I ask:
> > I suspect closed-form Cayley map is faster with $(c_{in}*h)\times (c_{out}*w)$ than $(c_{in})\times (c_{out}*h*w)$.
> > Does the difference not correspond to whether we must invert a $c_{out}*h*w \times c_{out}*h*w$ matrix or invert a $c_{in}*h\times c_{out}*w$ matrix?
> > Since inversion takes cubic time, the $(c_{in}*h) \times (c_{out}*w)$ reshaping is $w^3$ times faster, assuming $h=w$ and $c_{out}=c_{in}$.
> > In table 3 on page 7, it seems like your algorithm is around 5 times faster than the closed-form Cayley map.
> > If $w=3$ the closed-form Cayley map would be $3^3=27$ times faster, roughly 5 times faster than your algorithm on the $c_{in}\times (c_{out}*h*w)$ reshaping.
> >
> > Question: Is the gains of the particular reshaping really worth slowing down the closed-form Cayley map $27$ times?
> >
> > I guess there's probably something I'm not understanding right.

---

> > > ### Author Response · Authors · 2020-01-25
> > > **An interesting strategy for future exploration, but its meaning is not clear now.**
> > >
> > > Thank you for the question. We do not have the results for your reshaping. Your reshaping might be faster, but it would be valuable only when we can give it a good interpretation. Anyway, it's an interesting strategy of kernel reshaping for future exploration.

---

### Decision · Program_Chairs · 2019-12-19

**Decision:**

Accept (Poster)

**Comment:**

This paper presents a method for optimizing parameter matrices of deep learning objectives while enforcing orthonormality constraints.  While advantageous in certain respects, such constraints can be expensive to maintain when using existing methods.  To address this issue, an new algorithm is proposed based on the Cayley Transform and analyzed in terms of convergence.  After the discussion period two reviewers supported acceptance while one still voted for rejection.  Consequently, in recommending acceptance here for a poster, it is worth examining the significance of unresolved concerns.

First, the reject reviewer raised the valid point that the convergence proof relies on the assumption of Lipschitz continuous gradients, and yet the experiments use ReLU activation functions that do not satisfy this criteria.  In my view though, it is sometimes reasonable to derive useful theory under the assumption of Lipschitz continuous derivatives that nonetheless provides insight into the case where these derivatives may not be Lipschitz on a set of measure zero (which would be the case with ReLU activations).  So while ideally it might be nice to extend the theory to remove this assumption, the algorithm seems to work fine with ReLU activations in practice.  And this seems reasonable given the improbability of any iterate exactly hitting the measure-zero points where the gradients are discontinuous.  Beyond this issue, some criticisms were mentioned in terms of how and where the timing comparisons were presented.  However, I believe that these issues can be easily remedied in a final revision.